# The Role of PRMT5 in Immuno-Oncology

**DOI:** 10.3390/genes14030678

**Published:** 2023-03-09

**Authors:** Yoshinori Abe, Takumi Sano, Nobuyuki Tanaka

**Affiliations:** Department of Molecular Oncology, Institute for Advanced Medical Sciences, Nippon Medical School, 1-1-5 Sendagi, Bunkyo-Ku, Tokyo 113-8602, Japan

**Keywords:** PRMT5, immuno-oncology, immune evasion, immune checkpoint, tumor microenvironment

## Abstract

Immune checkpoint inhibitor (ICI) therapy has caused a paradigm shift in cancer therapeutic strategy. However, this therapy only benefits a subset of patients. The difference in responses to ICIs is believed to be dependent on cancer type and its tumor microenvironment (TME). The TME is favorable for cancer progression and metastasis and can also help cancer cells to evade immune attacks. To improve the response to ICIs, it is crucial to understand the mechanism of how the TME is maintained. Protein arginine methyltransferase 5 (PRMT5) di-methylates arginine residues in its substrates and has essential roles in the epigenetic regulation of gene expression, signal transduction, and the fidelity of mRNA splicing. Through these functions, PRMT5 can support cancer cell immune evasion. PRMT5 is necessary for regulatory T cell (Treg) functions and promotes cancer stemness and the epithelial–mesenchymal transition. Specific factors in the TME can help recruit Tregs, tumor-associated macrophages, and myeloid-derived suppressor cells into tumors. In addition, PRMT5 suppresses antigen presentation and the production of interferon and chemokines, which are necessary to recruit T cells into tumors. Overall, PRMT5 supports an immunosuppressive TME. Therefore, PRMT5 inhibition would help recover the immune cycle and enable the immune system-mediated elimination of cancer cells.

## 1. Introduction

The accumulation of genetic changes in cells can lead to cancer development. Cancer cells lose their normal regulatory mechanisms, resulting in increased cell proliferation and invasion rates [1,2]. Although the immune system can remove aberrant cells [3], some of these cells use various mechanisms to escape the immune attack. Such cancer cells can then exclusively proliferate, leading to tumor formation [4,5]. Through a well-known mechanism, cancer cells express certain factors that can suppress T cell function. Cancer cells are maintained within the tumor microenvironment (TME). The TME consists of various cell types, such as immunosuppressive cells, cancer-associated fibroblasts (CAFs), and cancer stem cells (CSCs). CSCs help recruit immunosuppressive cells, such as regulatory T cells (Tregs), tumor-associated macrophages (TAMs), and myeloid-derived suppressor cells (MDSCs), into the tumor [6]. A type of CAF (CAF-S1), which express α-smooth muscle actin (α-SMA) and fibroblast activation protein (FAP), also recruits Tregs into the tumor [7]. Therefore, restoring the anti-cancer abilities of immune cells is essential for successful cancer therapy. Using this concept, immune checkpoint inhibitors (ICIs) have been developed. ICIs recover the immune system’s ability to attack cancer through binding to immune checkpoint molecules, such as programmed death receptor-1 (PD-1), programmed death receptor ligand-1 (PD-L1), and cytotoxic T-lymphocyte antigen 4 (CTLA-4) [8]. Tregs, TAMs, and MDSCs express these immune checkpoint molecules. However, ICI treatment is effective for only a subset of cancer patients [9]. It is believed that the response to ICI treatment depends on the cancer type and its associated TME [10,11]. Therefore, various therapeutic strategies have been developed. For example, a combination treatment of a mitogen-activated protein kinase (MEK) inhibitor and an ICI proceeded to clinical trial [12,13].

Protein arginine methyltransferases (PRMTs) methylate arginine residues in their substrates. PRMTs regulate gene expression, signal transduction, and the fidelity of mRNA splicing [14]. Recently, it has been reported that PRMTs maintain Tregs, cancer stemness, and the epithelial–mesenchymal transition (EMT), which are critical factors for maintaining an immunosuppressive TME [15,16]. Among PRMTs, PRMT5 is believed to be highly associated with tumor formation, with PRMT5 overexpression observed in various cancers [17,18]. In this review article, we present the current evidence revealing that PRMT5 is involved in cancer immunology. PRMT5 suppresses cytokines and chemokine production in antigen-presenting cells (APCs), which can prevent the infiltration of T cells. Furthermore, PRMT5 has an important role in regulatory T cell (Treg) maintenance. Additionally, PRMT5 helps maintain the TME, which supports cancer cell survival and immune evasion. Therefore, inhibiting PRMT5 may help recover the cancer-immunity cycles, making it a potentially promising therapeutic target.

## 2. Immuno-Oncology

### 2.1. The Cancer-Immunity Cycle and Cancer Immunoediting

The concept of the cancer-immunity cycle has been proposed to represent the complex and dynamic mechanism used by the immune system to kill cancer cells (Figure 1) [3]. In Step 1, dead cancer cells produce cancer cell-specific antigens, also called neoantigens, which are captured by antigen-presenting cells (APCs), such as dendritic cells (DCs). Next, APCs present the captured antigens on major histocompatibility complexes I and II (MHC I and MHC II) to T cells (Step 2). Then, the effector T cell response against cancer-derived antigens is activated in the lymph node (Step 3). The activated effector T cells then migrate (Step 4) and infiltrate (Step 5) into the tumor. Finally, infiltrated effector T cells recognize cancer cells (Step 6) that express the specific antigens and attack them (Step 7). The cancer cells killed by these T cells then express new cancer cell-specific antigens (return to Step 1), beginning the cycle again. 

Although the human body has excellent immune system mechanisms against cancer cells, the cycle does not work properly in cancer patients. To explain this, the cancer immunoediting concept was proposed (Figure 2) [4,5]. When aberrant cells first appear in the body, immune cells, such as CD8+ T cells, macrophages, and natural killer (NK) cells, eliminate them. DCs capture aberrant cell-derived antigens and present them with MHCs to naïve T cells (elimination phase). Then, aberrant cells, which have low immunogenicity, evade the T cell-mediated attack. In this phase, in the presence of interleukin 12 (IL-12) and interferon, CD8+ T cells and CD4+ T cells eliminate the outgrowth of aberrant cells (equilibration phase). The cancer cells can evade the immune system by removing tumor antigens and MHCs and proliferating, leading to tumor formation (escape phase). 

### 2.2. Immune Checkpoint Machinery–A Key Factor for Evading the Cancer-Immunity Cycle

There are several factors that can regulate the cancer-immunity cycle through either promoting or repressing it. Cancer cells and the cancer-associated microenvironment can inhibit this cycle. In step 3 of the cancer-immunity cycle, antigen recognition through T cell receptors and the interaction of CD28 on T cells with CD80/CD86 on DCs are necessary for full T cell activation [19]. CTLA-4 exhibits a high affinity to CD80/CD86 because the extracellular domain structure of CTLA-4 is similar to that of CD28. Therefore, the interaction of CTLA-4 and CD80/CD86 inhibits the T cell activation signal, resulting in the suspension of the cancer-immunity cycle [20]. Vascular endothelial growth factor (VEGF), which is excessively produced by cancer cells, is also thought to suppress T cell infiltration (step 5 in the cancer-immunity cycle) [21]. Chronic cancer-antigen stimulation of effector T cells attenuates their activity, leading to the expression of immune checkpoint factors such as PD-1. Cancer cells also express interacting partners of factors that suppress T cell function, such as PD-L1. The interaction of PD-1 and PD-L1 allows cancer cells to evade immune attacks [22,23]. Furthermore, activated T cells express other checkpoint factors, such as lymphocyte-activation gene 3 (LAG-3) [24] and T cell immunoglobulin and mucin-domain containing-3 (TIM-3) [25], which also suppress T cell activity.

Several factors that contribute to the TME, such as CSCs, are also involved in evading the cancer-immunity cycle. These factors can support the recruitment of immunosuppressive cells into the TME (as described in Section 5). 

## 3. Overview of PRMT5 Functions

### 3.1. PRMT Family

The PRMT family is conserved from yeast to mammals and methylates specific arginine residues on target substrates. PRMTs transfer S-adenosylmethionine (SAM) to the guanidino nitrogen atoms of arginine, resulting in the formation of methylarginine (Figure 3) [14]. The mammalian PRMT family consists of nine PRMTs that are classified into three categories, depending on the arginine methylation process [26]. Type I PRMTs catalyze asymmetric di-methylation through monomethyl arginine. Type II PRMTs catalyze symmetric di-methylation through monomethyl arginine. Type III PRMT catalyzes only the monomethyl arginine. PRMTs contribute to various cellular regulatory mechanisms, such as gene transcription, mRNA splicing, signal transduction, DNA damage response, and cell fate decisions [26,27,28]. In this review, we focus on the role of a major type II PRMT, PRMT5, recognized as an anti-cancer target that has recently gained significant interest. In Section 5, we introduce the role of PRMT5 in immuno-oncology. Here, we briefly introduce the role of other PRMTs (coactivator-associated arginine methyltransferase 1 (CARM1)/PRMT4 and PRMT7) in immuno-oncology. Kumar et al. found that CARM1/PRMT4 inhibition could enhance anti-tumor immune system responses [29]. In T cells, CARM1/PRMT4 inactivation enhances their anti-tumor activity. Furthermore, in cancer cells, CARM1/PRMT4 inhibition upregulates type I interferon (IFN) and supports the infiltration of natural killer cells and CD8+ T cells into the tumor. PRMT7 inhibition also enhances IFN and chemokine production in melanoma cells [30]. 

### 3.2. Overview of PRMT5

PRMT5 was first identified as a binding protein of Janus kinase 2 (JAK2) [32]. PRMT5 expression is ubiquitous, and it is localized to the cytoplasm and nucleus. Among the PRMT5 interacting partners, methylosome protein 50 (MEP50)/WD repeat 77 (WDR77) is known as a major binding partner [33,34]. In vitro methylation assays revealed that the PRMT5–MEP50 complex methylates Sm proteins to a greater extent than PRMT5 alone, suggesting that MEP50 induces the full activity of PRMT5 [33]. Furthermore, MEP50 can bind to PRMT5 substrates, suggesting that MEP50 regulates the substrate specificity of PRMT5 [33,35]. PRMT5 activity is also regulated by other binding partners, including swelling-induced chloride conductance regulatory protein (pICln) and RIO kinase 1 (RIOK1) [36,37]. 

### 3.3. The Critical Role of PRMT5

#### 3.3.1. Epigenetic Regulation of Gene Expression

PRMT5-mediated epigenetic regulation of gene expression is regulated by the arginine methylation of histones H2, H3, and H4. PRMT5 mediates the dual symmetrical di-methylation of H2AR3 (H2AR3me2s) and H4R3 (H4R3me2s). This modification contributes to transcriptional repression by reducing the levels of H4K5ac [38]. In addition, PRMT5-mediated methylation of H3R8 (H3R8me2s) is also involved in transcriptional repression [39]. PRMT5-mediated methylation of H3R2 (H3R2me2s) is recognized by the WD40 domain of WD repeat 5 (WDR5) and promotes the trimethylation of H3K4 (H3K4me3) on chromatin (Figure 4a,b). This machinery induces transcriptional activation [40].

#### 3.3.2. mRNA Splicing

The regulation of mRNA splicing is a crucial role of cytoplasmic PRMT5. The mRNA splicing process is performed by the ribonuclear protein (RNP) complex (also known as the spliceosome), which contains small nuclear RNPs (snRNPs) and seven Sm proteins (B, D1, D2, D3, E, F, and G) [37]. PRMT5 methylates three out of the seven Sm proteins (B, D1, and D3). PRMT5-mediated methylation of Sm proteins enhances their affinity for the survival motor neurons (SMN) complex, which is important for splicing fidelity (Figure 4c) [33,37,41,42]. DBIRD is named as a protein complex of deleted in bladder cancer 1 (DBC1: also known as cell cycle and apoptosis regulator 2 (CCAR2)) and zinc finger 326 (ZNF326) with RNA polymerase II (Pol II). This complex may induce transcription elongation at the boundary of introns and AT-rich exons to promote their exclusion [43]. PRMT5-mediated methylation of ZNF326 is critical for the fidelity of alternative splicing at the boundaries of introns and AT-rich exons (Figure 4d) [44].

#### 3.3.3. Signal Transduction

Arginine methylation is also part of the regulation of signaling pathways. Examples include PRMT5-mediated arginine methylation of components of the epidermal growth factor receptor (EGFR), AKT, and hedgehog (HH) signaling pathways (Figure 4e).

PRMT5 di-methylates the intracellular domain of EGFR (Arg 1175). This modification induces the autophosphorylation of a tyrosine residue at residue 1173 and the binding of protein tyrosine phosphatase, Src homology region two domain-containing phosphatase 1 (SHP1). Then, the EGFR-SHP1 interaction attenuates the extracellular signal-regulated kinases (ERK) [45]. PRMT5-mediated methylation of rapidly accelerated fibrosarcoma (RAF) family members B-RAF and C-RAF induce their protein destabilization, resulting in the attenuation of ERK1/2 activity [46].

PRMT5-mediated AKT methylation regulates its activity and cellular localization. PRMT5 methylates AKT at arginine residues 15 and 391. These modifications promote AKT activation, leading to AKT membrane translocation and subsequent activation by phosphoinositide-dependent kinase-1 (PDK1) and the target of rapamycin complex 2 (mTORC2) [47].

PRMT5 also regulates the activity of transcriptional regulators. PRMT5 methylates arginine residues in the oligomerization domain of p53 [48]. PRMT5-mediated methylation of p53 affects the target gene specificity of p53. PRMT1 and PRMT5 methylate E2F1, and PRMT1 and PRMT5 have opposite roles for E2F1 methylation [49]. PRMT1-mediated methylation of E2F1 inhibits PRMT5-mediated methylation, which augments E2F1-dependent apoptosis. However, PRMT5-mediated methylation of E2F1 upregulates gene expression associated with cell cycle progression and cell survival. It has been reported that PRMT5 methylates a subunit of NF-kB, p65, which occurs by cytokine stimulation. IL-1β stimulation enhances PRMT5-mediated methylation of p65 arginine residue 30. This modification stabilizes p65 and promotes its DNA binding to kB elements, resulting in transcriptional activation [50]. Tumor necrosis factor α (TNF-α) stimulation also enhances PRMT5-mediated methylation of p65 arginine residues 30 and 35. In this case, PRMT5-mediated methylation of p65 enables its binding to the C-X-C motif chemokine ligand 10 (CXCL10) promoter, thus inducing its transcription [51].

The glioma-associated oncogene (GLI) family transcriptional regulator is activated downstream of the HH signaling pathway. PRMT5 methylates cytoplasmic GLI1 arginine residues 990 and 1018. These modifications induce GLI1 protein stabilization by preventing the interaction of GLI1 with the itchy E3 ubiquitin protein ligase (ITCH)-NUMB E3 ligase complex [35]. In addition, PRMT5 is involved with forkhead box P3 (FOXP3) in Tregs [52]. The role of this modification is described in Section 5.2.

## 4. The Role of PRMT5 in the Immune System

As described in the previous section, PRMT5 has essential roles in gene expression and mRNA splicing. It has been reported that these functions of PRMT5 have important roles in T cell maintenance and B cell development. Although PRMT5 inhibitors may attenuate T cell or B cell functions, side effects have not yet been reported. In addition, a previous report [53] did not mention that PRMT5 inhibition can affect the attenuation of the anti-tumor immune system (see Section 5.2).

### 4.1. Roles of PRMT5 in T Cell Maintenance

IL-2 is produced during the antigen-specific clonal growth of T cells and is essential for the proliferation and survival of CD4+ T cells and CD8+ T cells [54]. PRMT5 promotes IL-*2* gene expression after T cell signaling activation [55]. IL-2 and its family of cytokines (IL-4, IL-7, IL-9, IL-15, and IL-21) share the IL-2 receptor γ-chain (IL-2Rγ) as a receptor component, known as the common cytokine receptor γ chain (γc) [56]. These cytokines can activate JAK1 and JAK3, resulting in signal transducer and activator of transcription 5 (STAT5) transcriptional activity. Inoue et al. reported the mechanism of PRMT5-mediated T cell maintenance [15]. PRMT5 expression and its activity are upregulated in CD4+ and CD8+ T cells by stimulation with anti-CD3 and anti-CD28 antibodies. The upregulation of PRMT5 is a critical factor for invariant natural killer T cell (iNKT cell), CD4+ T cell, and CD8+ T cell proliferation and survival. Mechanistically, PRMT5 regulates γc and JAK3 expression through precise mRNA splicing of those genes by PRMT5-mediated methylation of the Sm protein SmD3. Loss of PRMT5 can abolish T cell proliferation because of failed STAT5 activation [15]. 

### 4.2. Roles of PRMT5 in B Cell Development

B cells are part of the adaptive immune system and produce high-affinity antibodies that mediate protection from pathogens, but remain tolerant of self-tissues. At first, B cell progenitor cells are generated in the bone marrow. Then, progenitor cells transform into mature B cells. In this stage, progenitor cells repeatedly proliferate and differentiate to undergo successful rearrangement of the immunoglobulin genes. Activated mature B cells enter the germinal center (GC). Then, they undergo expansion, immunoglobulin class switching, and programmed Ig mutation coupled to antibody affinity-based selection. Finally, mature B cells differentiate into memory B cells or plasma cells [57].

Litzler et al. found that PRMT5 is necessary for B cell development [16]. PRMT5 mRNA expression levels are elevated in the Pro-B cells, which involve the rearrangement of the immunoglobulin gene and express immunoglobulin α and β (Igα and Igβ) heterodimers, and early Pre-B cell stages that express pre-B cell receptors (preBCR), as well as activated B cells. Furthermore, immunohistochemical analysis revealed that PRMT5 protein levels are elevated in B cells localized in GCs. PRMT5 is involved in B cell proliferation, survival, and protection from apoptosis in GCs. Furthermore, PRMT5 is necessary for antibody responses through promoting GC formation. As described in Section 2.2, PRMT5 regulates gene expression and mRNA splicing. RNA-sequencing analysis comparing normal B cells and B cells with knocked out PRMT5 revealed that PRMT5 downregulates pro-apoptotic genes and cell cycle inhibitor genes in response to p53 in B cell development. Furthermore, PRMT5 is involved in the fidelity of mRNA splicing for murine double minute protein (MDM4) [58], a repressor of p53 and subunits of Tat interactive protein 60 kDa (TIP60), complex histone deacetylase 6 (HDAC6) and cadherin 1 (CDH1), which regulates p53-mediated DNA repair [59,60]. However, the loss of PRMT5 can also induce p53-independent apoptosis.

## 5. The Roles of PRMT5 in Immuno-Oncology and Implication for Cancer Therapy Targeting PRMT5

PRMT5 overexpression has been observed in several cancers (see Section 5.1). Furthermore, emerging evidence shows that PRMT5 is involved in immune evasion in cancer. In this section, we introduce the association between PRMT5 and clinical relevance, especially in immuno-oncology. A schematic illustration is represented in Figure 5.

### 5.1. PRMT5 as a Cancer Therapy Target 

The overexpression of PRMT5 is observed in various cancers and is associated with worse survival rates [17,61,62,63,64,65]. The effect of PRMT5 overexpression in cancers is summarized in Table 1.

PRMT5-mediated histone modification is involved in CSC survival, and deregulated PRMT5 mRNA splicing machinery can induce strange insertions of introns and exon skipping that result in gene mutations. Furthermore, PRMT5 upregulates the expression of various genes that are associated with oncogenic signaling, CSC renewal, metastasis, and metabolic reprogramming [61]. Recently, it has been reported that PRMT5 is associated with immune surveillance [52]. Therefore, PRMT5 is an attractive target for anti-cancer drug development. The first PRMT5 inhibitor (GSK3235025/EPZ015666) was developed and was effective for lymphoma [66]. Subsequently, various PRMT5 inhibitors have been developed, and some have undergone phase I (JNJ-64619178 and PF0693999) or phase II (GSK3326595) clinical trials [62].

Homozygous deletion of methylthioadenosine phosphorylase (MTAP) frequently occurs in cancer because of its proximity to the *CDKN2A* gene, which encodes p16/INK4A and p19/ARF [67]. In addition, previous reports have shown that MTAP loss attenuates PRMT5 activity. Furthermore, the PRMT5 inhibitor GSK3230591 is more sensitive in some cancer cells that do not express MTAP [68,69].

### 5.2. PRMT5-Mediated cGAS-STING Pathway

The stimulator of the interferon gene (STING) is a transmembrane protein in the endoplasmic reticulum that is activated by cytosolic double-stranded DNA (dsDNA). Cyclic GMP-AMP synthase (cGAS) acts as a sensor for dsDNA and is necessary for STING activation. Activated STING upregulates the interferon regulating factor 3 (IRF3)-TANK binding kinase 1 (TBK1) cascade, resulting in the production of cytokines and type I interferon [70].

Kim et al. reported that PRMT5 can attenuate the cGAS-STING pathway, which supports immune evasion in melanoma [53]. PRMT5 symmetrically di-methylates interferon-gamma inducible protein 16 (IFI16), a cGAS complex component [71]. PRMT5-mediated methylation of IFI16 attenuates dsDNA-induced type I interferon and chemokine production. Therefore, PRMT5 suppresses the invasion of CD4+ T cells and CD8+ T cells into tumors. Furthermore, PRMT5 also suppresses the NLR family CARD domain containing 5 (NLRC5), inhibiting gene expression associated with MHC I antigen presentation. Therefore PRMT5-mediated suppression of antigen presentation also supports immune evasion in cancer (Figure 5①).

The authors also demonstrated the correlation between PRMT5 and the anti-tumor immune response by in vivo experiments. PRMT5 knockdown melanoma cell xenograft tumor growth was inhibited in immunocompetent C57BL/6 mice. However, PRMT5 knockdown did not affect tumor shrinkage in immunocompromised NOD-SCID gamma (NSG) mice. These results suggest that PRMT5 inhibition upregulates the anti-tumor immune response. Furthermore, the melanoma cell xenografted tumor was shrunk following combination treatment with a PRMT5 inhibitor (GSK3326595) [58] and an anti-PD-1 antibody.

### 5.3. PRMT5-Mediated Treg Maintenance and Cancer

Treg cells are essential for maintaining self-tolerance and immune homeostasis under normal conditions. They can also infiltrate into tumor tissues and hamper anti-tumor immune responses [72]. A member of the forkhead family transcription factor, FOXP3, has essential roles in Treg development and maintenance [73,74]. In human tumors, the infiltration of tumor sites with FOXP3+ Tregs is associated with a poor prognosis [75].

Nagai et al. found that PRMT5 can regulate Treg function through FOXP3 methylation [52]. PRMT5 was identified as a binding partner of FOXP3 and can symmetrically di-methylate it at arginine residues 48 and 51. PRMT5-mediated methylation of FOXP3 is necessary for its activation to maintain Treg function (Figure 5②). Mutant FOXP3 attenuates Treg suppressive function. Furthermore, FOXP3-mediated gene expression, associated with the cell cycle and inflammation, was altered in PRMT5-deleted T cells.

Cancer cell xenograft tumor analysis revealed that PRMT5 inhibitor (DS-437) [76] treatment shrunk the tumor mass. Furthermore, for the v-erb-b2 avian erythroblastic leukemia viral oncogene homolog 2 (ErbB2)/human epidermal growth factor receptor 2 (HER2) antibody (trastuzumab)-resistant cancer cells (CT26-her2 cells) xenografted tumor, a combination treatment of the PRMT5 inhibitor and Trastuzumab dramatically reduced tumor growth. Mechanistically, this combinational treatment induces infiltrating CD8+ T cells and NK cells into the tumor tissue and inhibits Treg function.

### 5.4. PRMT5-Mediated CSC Maintenance

CSCs are a subpopulation of cancer cells that exhibit the characteristics of both stem cells and tumor cells. CSCs drive tumor initiation and self-renewal. CSCs have enhanced drug-efflux pumps, a high DNA repair capacity, and upregulated protection against reactive oxygen species (ROS). Therefore, chemotherapy and radiation therapy have minimal effects on these cells [77]. CSCs can produce cytokines, chemokines, and growth factors that help them to evade the immune surveillance system. IL-1β and IL-6 released from CSCs can recruit MDSCs. C-C motif chemokine ligand 1 (CCL1), CCL2, CCL5, and CXCL5 recruit MDSCs and Tregs [6]. Transforming growth factor-β (TGF-β) secreted from CSCs induces Treg differentiation [78]. In addition, CSCs highly express PD-L1, which is involved in evading T cell attacks [79,80].

Accumulating evidence shows that suppressing PRMT5 can attenuate the CSC population in tumors (Figure 5③). PRMT5 induces H3R2me2s at CSC-associated gene promoters, leading to transcriptional activation (see also Section 3.3.1) [40]. Using this common mechanism, H3R2me2 regulates stemness-associated gene expression, which is dependent on tumor type. In breast CSCs, PRMT5 epigenetically upregulates the expression of FOXP1 [81]. In leukemia CSCs, PRMT5 drives the expression of disheveled segment polarity protein 3 (DVL3), an upstream positive regulator of WNT/β-catenin signaling [82]. Furthermore, PRMT5 is also involved in glioblastoma CSC maintenance through mRNA splicing fidelity, affecting cell cycle regulators [51]. Although not included in this article, PRMT5 also regulates cancer cell proliferation [83] (Figure 5③).

### 5.5. PRMT5-Mediated EMT

Cancer cells can spread from the primary tumor site to nearby tissues, and in some cases to distant tissues, forming a new tumor. This process is called metastasis. EMT is essential for tumor metastasis [84]. EMT is induced by activating various signaling pathways, including the TGF-β, EGF, hepatic growth factor (HGF), NOTCH, or HH signaling pathways [85,86,87,88]. In addition, EMT-associated genes also suppress the infiltration of CD8+ T cells [89,90] and promote Treg, MDSC, and TAM infiltration. Furthermore, EMT-induced cancer cells have low MHC expression levels and high PD-L1 expression levels [91]. In this section, we introduce some previous reports showing that PRMT5 is associated with EMT.

AKT acts downstream of TNF-α and TGF-β to upregulate EMT-associated transcription factors, such as SNAIL and TWIST1 [92,93,94]. In neuroblastoma, PRMT5 can di-methylate AKT1 arginine residue 15, leading to its activation. The PRMT5–AKT1 axis regulates the expression of SNAIL, TWIST1, and zinc-finger-enhancer binding protein 1 (ZEB1) [95]. 

Xenografted tumor analysis revealed that PRMT5 inhibitor treatment can attenuate tumor growth and metastasis. The PRMT5–AKT1 axis is also associated with EMT in lung and colorectal cancer [47].

Other previous reports have shown that PRMT5 forms a complex with SNAIL and nucleosome remodeling and deacetylase (NuRD) [96] or with ZEB2, TWIST1, and NuRD [97], and regulates EMT-associated gene expression. In cervical cancer, the PRMT5–SNAIL–NuRD complex suppresses *cadherin 1* (*CDH1*: also known as *E-cadherin*) and ten–eleven translocation methylcytosine dioxygenase 1 (*TET1*) gene expression by binding to their promoter regions [96]. The PRMT5–ZEB2–TWIST–NuRD complex also binds to the *CDH1* promoter and suppresses its gene expression in colon cancer [97].

In laryngeal carcinoma, PRMT5 regulates WNT4 expression, and PRMT5–WNT4 activates β-catenin to induce *cadherin 2* (*CDH2*: also known as *N-cadherin*) and *SNAIL* expression, which is involved in EMT and promotes cell migration and lymph-node metastasis [65] (Figure 5④).

## 6. Conclusions

The development of ICIs has brought a paradigm shift to cancer therapeutic development strategies. However, the response rate of ICIs remains low. Accordingly, the establishment of biomarkers can enable improvement in ICI efficiency. In addition, developing a combination therapy approach that uses molecular target drugs and ICIs would be an attractive cancer therapy strategy. PRMT5 overexpression is observed in a wide variety of cancers and is involved in the maintenance of the TME. PRMT5 inhibition would disrupt an immunosuppressive TME and promote T cell infiltration. In this context, the combination of ICIs and PRMT5 treatment would be another therapeutic strategy. However, PRMT5 is involved in T cell maintenance and B cell development. For PRMT5 inhibition only in the TME, it is possible that PRMT5 expression levels and MTAP status (MTAP loss or not) would be essential factors for this approach.

## Figures and Tables

**Figure 1 genes-14-00678-f001:**
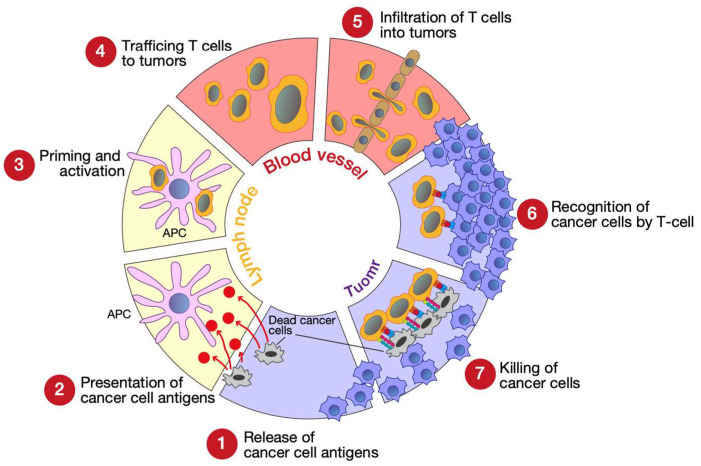
Seven steps of the cancer-immunity cycle. APC, antigen-presenting cell (e.g., dendritic cell). This figure was illustrated using Adobe Illustrator 2022 (Adobe, San Jose, CA, USA) based on ref. [3].

**Figure 2 genes-14-00678-f002:**
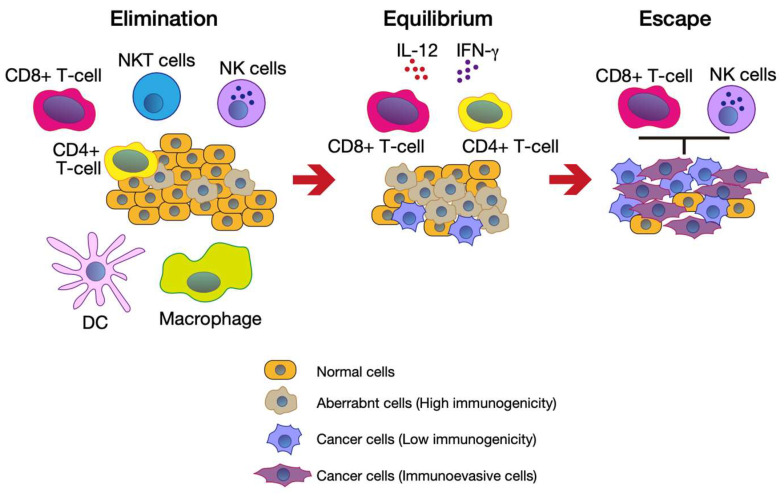
Cancer immunoediting. NKT cells, natural killer T cells; NK cells, natural killer cells; DC, dendritic cells. This figure was illustrated using Adobe Illustrator 2022 based on ref. [4].

**Figure 3 genes-14-00678-f003:**
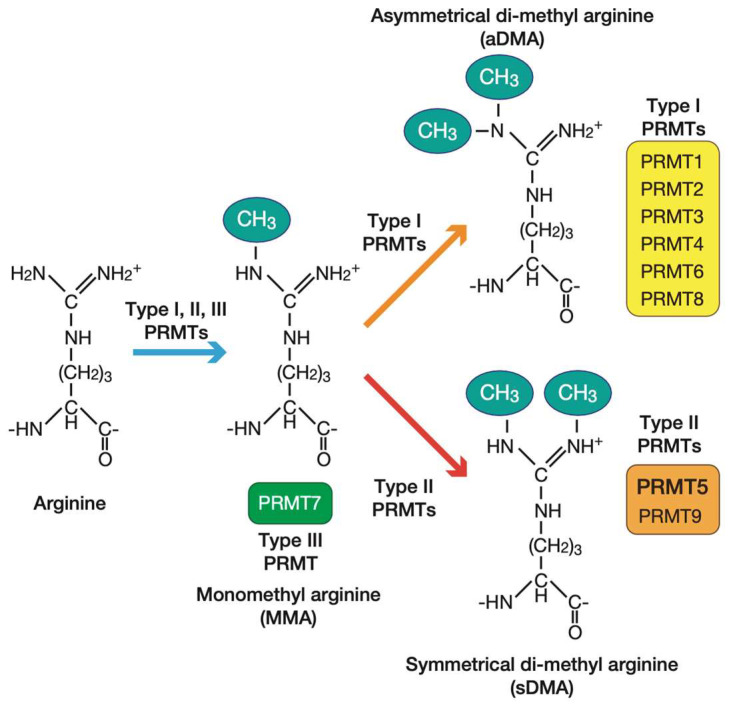
Protein arginine methyltransferase (PRMT)-mediated arginine methylation. All PRMT family proteins modify monomethylated arginine. Next, type I PRMTs modify asymmetrical methylated arginine and type II PRMTs modify symmetrically methylated arginine, while type III PRMT (PRMT7) modifies only monomethyl arginine. This figure was illustrated using Adobe Illustrator 2022 and was modified from our previous work [31].

**Figure 4 genes-14-00678-f004:**
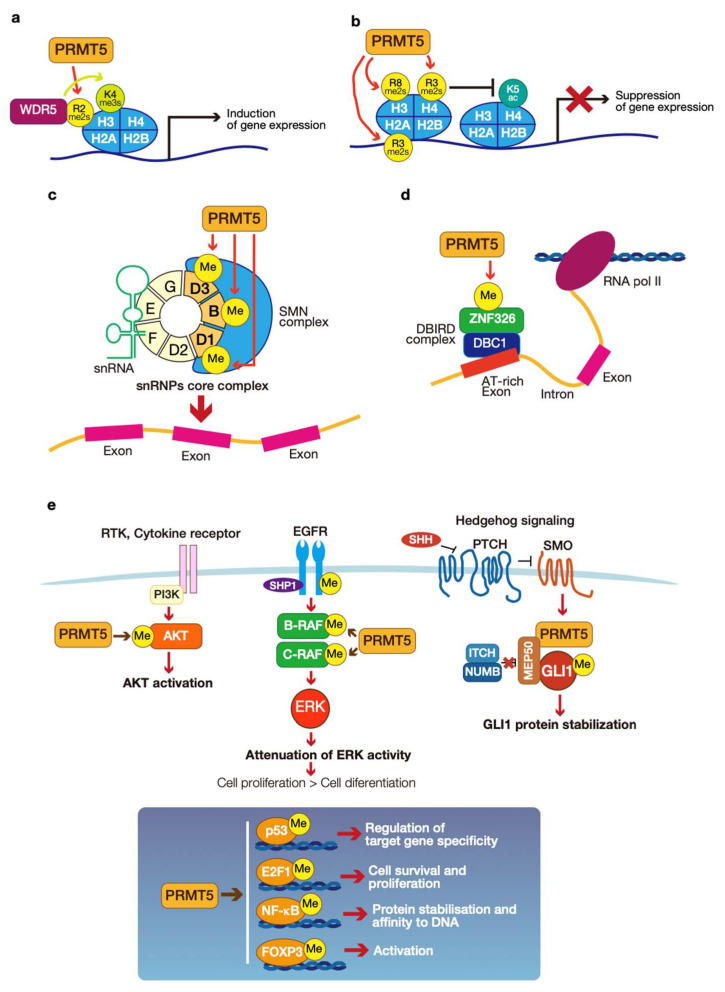
The roles of protein arginine methyltransferase 5 (PRMT5). (**a**,**b**): Schematic illustrations of PRMT5-mediated epigenetic regulation of gene expression. (**c**,**d**): The PRMT5-meditated mRNA splicing machinery. (**e**): PRMT5-mediated signal transduction and transcriptional regulatory molecules. Each figure is the author’s original image, illustrated using Adobe Illustrator 2022.

**Figure 5 genes-14-00678-f005:**
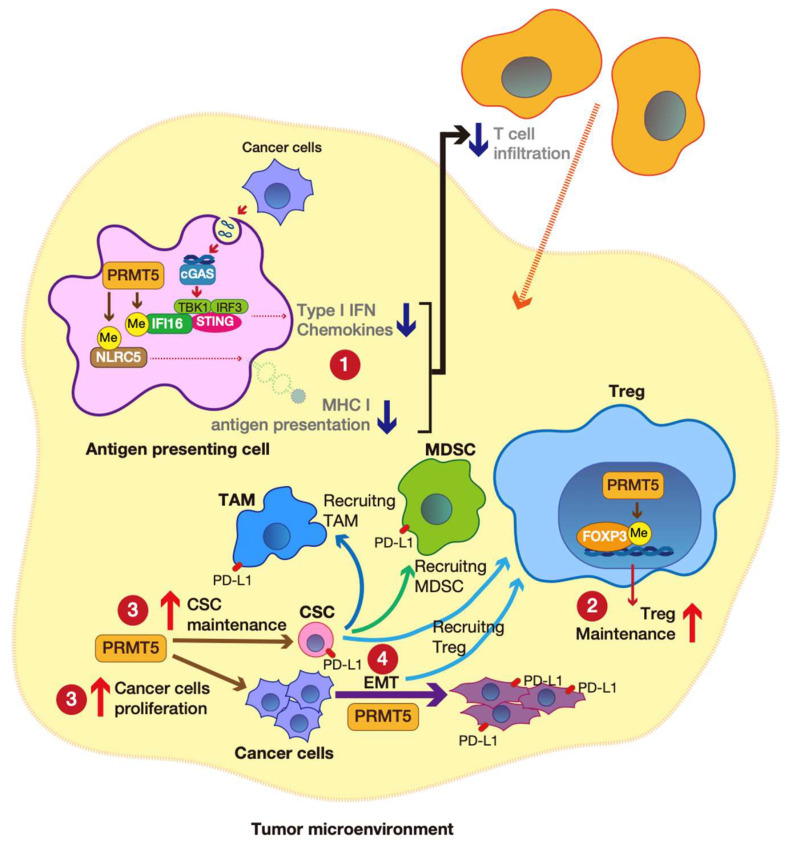
The roles of protein arginine methyltransferase 5 (PRMT5) in immune evasion. PRMT5 attenuates type I interferon (IFN) and chemokine production and represses antigen presentation, suppressing cancer cell recognition by T cells and infiltration of T cells into the tumor ①. PRMT5 activates forkhead box P3 (FOXP3), which is essential for regulatory T cell (Treg) maintenance ②. PRMT5 maintains cancer stemness and cancer cell proliferation ③. PRMT5 upregulates epithelial-to-mesenchymal transition (EMT)-associated gene expression, leading to metastasis ④. CSCs and EMT promote regulatory T cell (Treg), tumor-associated macrophage (TAM), and myeloid-derived suppressor cell (MDSC) recruitment into the tumor microenvironment (TME), supporting immune evasion. This figure is the author’s original image, illustrated using Adobe Illustrator 2022.

**Table 1 genes-14-00678-t001:** The role of PRMT5 overexpression in cancer.

**PRMT5 Overexpression**	**Effect of PRMT5 Overexpression**
Lung cancer	Larger tumor size, advanced tumor grade, lymph node metastasis, worse survival
Breast cancer	Larger tumor size, advanced tumor grade, lymph node metastasis
Hepatocellular carcinoma	Larger tumor size, advanced tumor grade
Glioblastoma multiforme	Larger tumor size, advanced tumor grade
Pancreatic cancer	Worse survival
Laryngeal carcinoma	Worse survival
Gastric cancer	Lymph node metastasis
Head and neck cancer	Lymph node metastasis
Colorectal cancer	Lymph node metastasis
Ovarian caner	Lymph node metastasis
Bladder cancer	Lymph node metastasis
Melanoma	Metastasis

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
