# Peer review of "The Role of PRMT5 in Immuno-Oncology"

_genes, 2023, doi:10.3390/genes14030678_

Round 1

Reviewer 1 Report

In this review, Abe et al. comprehensively summarized the role of PRMT5 in Immuno-Oncology. Overall, the paper is well-organized. 

1.     Section 2-2: Evading the Cancer-Immunity Cycle. It’s a big title involving multiple subjects, such as immune checkpoint, downregulation of MHC class I, immune suppressive factors (e.g. TGFb), immune suppressive cells (e.g. MDSC, TAM, Treg, CAFs..), etc. However, the authors only focused on immune checkpoint in this section, which is rather limited.

2.     Section 3-1. PRMT Family. It would be helpful to discuss the roles of the other PRMT family members in cancer immunology besides PRMT5. For example, recent study showed that CARM1 inhibition enables immunotherapy of resistant tumors by dual action on tumor cells and T Cells (PMID: 33707234).

3.     The authors claimed that PRMT5 is necessary for T cell proliferation and B cell development. Would this cause severe side effects when using PRMT5 inhibitors in anti-cancer therapy?

4.     It would be helpful to discuss the relevance of PRMT5 in human cancer. For example, it’s expression level in different tumors compared with normal tissue, or its correlation with survival rates of patients with cancer.

Author Response

Reply to reviewer 1

  1. Section 2-2: Evading the Cancer-Immunity Cycle. It’s a big title involving multiple subjects, such as immune checkpoint, downregulation of MHC class I, immune suppressive factors (e.g. TGFb), immune suppressive cells (e.g. MDSC, TAM, Treg, CAFs..), etc. However, the authors only focused on immune checkpoint in this section, which is rather limited.

Thank you for your helpful comment. We have changed the title of section 2-2 to “Immune Checkpoint machinery – a Key Factor for Evading the Cancer-Immunity Cycle”.

  1. Section 3-1. PRMT Family. It would be helpful to discuss the roles of the other PRMT family members in cancer immunology besides PRMT5. For example, recent study showed that CARM1 inhibition enables immunotherapy of resistant tumors by dual action on tumor cells and T Cells (PMID: 33707234).

Thank you for the helpful suggestion. We have added information from recent studies that focused on the role of CARM1 or PRMT7 in cancer immunology, as described below (Lines 124–131 in the revised manuscript).

“In section 5, we introduce the role of PRMT5 in immuno-oncology. Here, we briefly introduce the role of other PRMTs (coactivator-associated arginine methyltransferase 1 (CARM1)/PRMT4 and PRMT7) in immuno-oncology. Kumar et al. found that CARM1/PRMT4 inhibition could enhance anti-tumor immune system responses [29]. In T cells, CARM1/PRMT4 inactivation enhances their anti-tumor activity. Furthermore, in cancer cells, CARM1/PRMT4 inhibition upregulates type I interferon (IFN) and supports the infiltration of natural killer cells and CD8+ T cells into the tumor. PRMT7 inhibition also enhances IFN and chemokine production in melanoma cells [30].”

  1. The authors claimed that PRMT5 is necessary for T cell proliferation and B cell development. Would this cause severe side effects when using PRMT5 inhibitors in anti-cancer therapy?

Thank you for your question. Although it is possible that PRMT5 inhibitors may attenuate T cell or B cell functions, side effects have not been reported yet. In addition, Kim et al. did not mention that PRMT5 inhibition affects the attenuation of the anti-tumor immune system (ref. 53 in the revised manuscript). Therefore, we have added the explanation included below (lines 214–217 in the revised manuscript).

“Although PRMT5 inhibitors may attenuate T cell or B cell functions, side effects have not been reported yet. In addition, a previous report [53] did not mention that PRMT5 inhibition can affect the attenuation of the anti-tumor immune system (see section 5-2).”

  1. It would be helpful to discuss the relevance of PRMT5 in human cancer. For example, it’s expression level in different tumors compared with normal tissue, or its correlation with survival rates of patients with cancer.

Thank you for the helpful advice. We have summarized the cancer types in which PRMT5 overexpression was observed and the resulting effects in Table 1.

Reviewer 2 Report

The manuscript entitled” The Role of PRMT5 in Immuno-Oncology” by Abe et al is a very good review article suggesting the comprehensive role of PRMT5 in cancer progression. The work also cited the concerned literature well.

1)      Please make brief sentences. Some sentences go beyond 3to 4 lines. I request the authors to look at the manuscript themselves and make shorter sentences.

2)      Please ensure that the diagrams are originally made by you and are not taken from elsewhere. Also, provide the name of the software that was used to make the diagrams.

Author Response

Reply to reviewer 2

1)      Please make brief sentences. Some sentences go beyond 3 to 4 lines. I request the authors to look at the manuscript themselves and make shorter sentences.

Thank you for your helpful suggestion. We restructured some of the sentences to be shorter, as described below. Line numbers in original manuscript were shown.

Lines 33–35: one sentence was divided into two sentences.

[Original] Cancer cells are maintained within the tumor microenvironment (TME), which consists of various cell types, such as immunosuppressive cells, cancer-associated fibroblasts (CAFs), and cancer stem cells (CSCs).

[Revised] Cancer cells are maintained within the tumor microenvironment (TME). The TME consists of various cell types, such as immunosuppressive cells, cancer-associated fibroblasts (CAFs), and cancer stem cells (CSCs).

Lines 42–45: Some words were deleted to make a shorter sentence.

[Original] ICIs help recover the immune system’s ability to attack the cancer by suppressing immunosuppressive signaling through binding to immune checkpoint molecules such as programmed death receptor-1 (PD-1), programmed death receptor ligand-1 (PD-L1), and cytotoxic T-lymphocyte antigen 4 (CTLA-4).

[Revised] ICIs help recover the immune system’s ability to attack cancer through binding to immune checkpoint molecules such asprogrammed death receptor-1 (PD-1), programmed death receptor ligand-1 (PD-L1), and cytotoxic T-lymphocyte antigen 4 (CTLA-4).

Lines 71–73: Some words were deleted to make a shorter sentence.

[Original] Then, the effector T cell response against cancer-derived antigen is activated in the lymph node (Step 3), after which activated effector T cells migrate (Step 4) and infiltrate (Step 5) into the tumor.

[Revised] Then, the effector T cell response against cancer-derived antigen is activated in the lymph node (Step 3). The activated effector T cells then migrate (Step 4) and infiltrate (Step 5) into the tumor.

Lines 96–98: Some words were deleted to make a shorter sentence.

[Original] CTLA-4, which is highly expressed in Tregs, exhibits high affinity to CD80/CD86 because the extracellular domain structure of CTLA-4 is similar to that of CD28.

[Revised] CTLA-4 exhibits high affinity to CD80/CD86 because the extracellular domain structure of CTLA-4 is similar to that of CD28.

Lines 99-103: Unnecessary information in this context was deleted to make a shorter sentence.

[Original] Vascular endothelial growth factor (VEGF), which is excessively produced by cancer cells, suppresses intercellular adhesion molecule 1 (ICAM1) expression, induces aberrant angiogenesis, and is thought to suppress T cell infiltration (step 5 in the cancer-immunity cycle).

[Revised] Vascular endothelial growth factor (VEGF) is also thought to suppress T cell infiltration (step 5 in the cancer-immunity cycle).

Lines 119–124: One sentence was divided into two sentences, and the name of the PRMT family molecules was deleted)

[Original] Type I PRMTs (PRMT1, PRMT2, PRMT3, coactivator-associated arginine methyltransferase 1 (CARM1)/PRMT4, PRMT6, and PRMT8) catalyze the asymmetric di-methylation through monomethyl arginine, and Type II PRMTs (PRMT5 and PRMT9) catalyze symmetric di-methylation through monomethyl arginine. Type III PRMT (PRMT7) cat-alyzes only the monomethyl arginine.

[Revised] Type I PRMTs catalyze the asymmetric di-methylation through monomethyl arginine. Type II PRMTs catalyze symmetric di-methylation through monomethyl arginine. Type III PRMT  catalyzes only the monomethyl arginine.

Lines 139–141: Some words were deleted to make shorter sentence.

[Original] Furthermore, MEP50 can bind to PRMT5 substrates, such as Sm proteins and GLI1, suggesting that MEP50 regulates the substrate specificity of PRMT5.

[Revised] Furthermore, MEP50 can bind to PRMT5 substrates, suggesting that MEP50 regulates the substrate specificity of PRMT5.

Lines 150–153: Unnecessary information was deleted, and a short sentence are added.

[Original] PRMT5-mediated symmetrical methylation of H3R2 (H3R2me2s) is recognized by the WD40 domain of WD repeat 5 (WDR5) and promotes recruitment of the mixed-lineage leukemia (MLL) complex and trimethylation of H3K4 (H3K4me3) on chromatin (Figure 4a, b).

[Revised] PRMT5-mediated methylation of H3R2 (H3R2me2s) is recognized by the WD40 domain of WD repeat 5 (WDR5) and promotes trimethylation of H3K4 (H3K4me3) on chromatin (Figure 4a, b). This machinery induces transcriptional activation.

Lines 159–161: Some words were deleted to make shorter sentences.

[Original] PRMT5-mediated methylation of Sm proteins enhances their affinity for the survival motor neurons (SMN) complex, which is important for functional spliceosome assembly, resulting in the maintenance of splicing fidelity (Figure 4c).

[Revised] PRMT5-mediated methylation of Sm proteins enhances their affinity for the survival motor neurons (SMN) complex, which is important for splicing fidelity (Figure 4c).

Lines 161–165: One sentence was divided into two sentences.

[Original] DBIRD is named as a protein complex of Deleted in bladder cancer 1 (DBC1: also known as cell cycle and apoptosis regulator 2 [CCAR2]) and zinc finger 326 (ZNF326) with RNA polymerase II (Pol II), which may induce transcription elongation at the boundary of introns and AT-rich exons to promote their exclusion.

[Revised] DBIRD is named as a protein complex of Deleted in bladder cancer 1 (DBC1: also known as cell cycle and apoptosis regulator 2 [CCAR2]) and zinc finger 326 (ZNF326) with RNA polymerase II (Pol II). This complex may induce transcription elongation at the boundary of introns and AT-rich exons to promote their exclusion.

Lines 189–191: One sentence was divided into two sentences.

[Original] However, PRMT5-mediated methylation of E2F1 upregulates gene expression associated with cell cycle progression and cell survival—PRMT5-mediated methylation of a subunit of NF-kB, p65, by cytokine stimulation.

[Revised] However, PRMT5-mediated methylation of E2F1 upregulates gene expression associated with cell cycle progression and cell survival. PRMT5-mediated methylation of a subunit of NF-kB, p65, occurs by cytokine stimulation.

Lines 199–201: One sentence was divided into two sentences.

[Revised] PRMT5 methylates cytoplasmic GLI1 arginine residues 990 and 1018. These modifications induce GLI1 protein stabilization by preventing the interaction of GLI1 with the itchy E3 ubiquitin protein ligase (ITCH)-NUMB E3 ligase complex.

Lines 213–215: Some words were deleted to make shorter sentence.

[Original] Furthermore, PRMT5 upregulates cancer-associated factors, such as oncogenic signaling genes, stem cell renewal genes, metastasis-associated genes, and metabolic reprogramming genes in cancer.

[Revised] Furthermore, PRMT5 upregulates expression of various genes that are associated with oncogenic signaling, CSC renewal, metastasis, and metabolic reprogramming in cancer.

Lines 218–221: Some words were deleted to make shorter sentence.

[Original] Subsequently, various PRMT5 inhibitors have been developed, and some have undergone phase I clinical trials, such as JNJ-64619178 and PF0693999, or phase II clinical trials, such as GSK3326595.

[Revised] Subsequently, various PRMT5 inhibitors have been developed, and some have undergone phase I (JNJ-64619178 and PF0693999) or phase II (GSK3326595) clinical trials.

Lines 239–241: Some words were deleted to make shorter sentence.

[Original] PRMT5 expression levels and arginine methyltransferase activity are upregulated in CD4+ and CD8+ T cells by stimulation with anti-CD3 and anti-CD28 antibodies.

[Revised] PRMT5 expression and its activity are upregulated in CD4+ and CD8+ T cells by stimulation with anti-CD3 and anti-CD28 antibodies.

Lines 242–245: Some words were deleted to make shorter sentence.

[Original] Mechanistically, PRMT5 regulates γc and JAK3 expression through precise mRNA splicing of those genes by PRMT5-mediated methylation of Sm protein SmD3, which is a component of the spliceosome.

[Revised] Mechanistically, PRMT5 regulates γc and JAK3 expression through precise mRNA splicing of those genes by PRMT5-mediated methylation of Sm protein SmD3.

Lines 249–252: One sentence was divided into three sentences.

[Original] B cell progenitor cells are generated in the bone marrow and transform into mature B cells, repeating proliferation and differentiation to undergo successful rearrangement of the immunoglobulin genes.

[Revised] At first, B cell progenitor cells are generated in the bone marrow. The progenitor cells then transform into mature B cells. In this stage, progenitor cells repeatedly proliferate and differentiate to undergo successful rearrangement of the immunoglobulin genes.

Lines 249–252: One sentence was divided into three sentences.

[Original] Activated mature B cells enter the germinal center (GC) and undergo expansion, immunoglobulin class switching, programmed Ig mutation coupled to antibody affinity-based selection, and differentiation into memory B cells or plasma cells.

[Revised] Activated mature B cells enter the germinal center (GC). Then, they undergo expansion, immunoglobulin class switching, and programmed Ig mutation coupled to antibody affinity-based selection. Finally, mature B cells differentiate into memory B cells or plasma cells.

Lines 294–297: The sentence were rewritten to make shorter sentences.

[Original] The authors also demonstrated the correlation between PRMT5 and anti-tumor immune response by in vivo experiment. PRMT5 knockdown melanoma cells-xenografted tumor growth was inhibited in immunocompetent C57BL/6 mice. However, PRMT5 knockdown did not affect tumor shrinkage in immunocompromised NOD-SCID gamma (NSG) mice. These results suggest that PRMT5 inhibition upregulates the anti-tumor immune response.

[Revised] The authors also demonstrated the correlation between PRMT5 and anti-tumor immune response by in vivo experiments. PRMT5 knockdown melanoma cell xenograft tumor growth was inhibited in immunocompetent C57BL/6 mice. However, PRMT5 knockdown did not affect tumor shrinkage in immunocompromised NOD-SCID gamma (NSG) mice. These results suggest that PRMT5 inhibition upregulates the anti-tumor immune response.

Lines 321–324: One sentence was divided into two sentences.

[Original] Because CSCs have enhanced drug-efflux pumps, high DNA repair capacity, and upregulated protection against reactive oxygen species (ROS), chemotherapy and radiation therapy have little effect on killing them.

[Revised] CSCs have enhanced drug-efflux pumps, high DNA repair capacity, and upregulated protection against reactive oxygen species (ROS). Therefore, chemotherapy and radiation therapy have little effect on killing them.

Lines 325–328: Some words were deleted, and one sentence was divided into two sentences.

[Original] IL-1β and IL-6 released from CSCs can recruit MDSCs and chemokines such as C-C motif chemokine ligand 1 (CCL1), CCL2, CCL5, and CXCL5 recruit MDSCs and Tregs, resulting in the formation of a TME that supports immune evasion.

[Revised] IL-1β and IL-6 released from CSCs can recruit MDSCs. C-C motif chemokine ligand 1 (CCL1), CCL2, CCL5, and CXCL5 recruit MDSCs and Tregs.

Lines 332–334: Some words were deleted to make shorter sentence.

[Original] PRMT5 induces H3R2me2s at CSC-associated gene promoters, leading to the recruitment of the WDR5-SET1 complex and trimethylation of H3K4 (H3K4me3s).

[Revised] PRMT5 induces H3R2me2s at CSC-associated gene promoters, leading to transcriptional activation (see also section 3-3-1).

Lines 359–361: One sentence was divided into two sentences.

[Original] Other previous reports have shown that PRMT5 forms a complex with SNAIL and nucleosome remodeling and deacetylase (NuRD) [91] or with ZEB2, TWIST1, and NuRD [92], and regulates EMT-associated gene expression.

[Revised] Other previous reports have shown that PRMT5 forms a complex with SNAIL and nucleosome remodeling and deacetylase (NuRD) [91] or with ZEB2, TWIST1, and NuRD [92]. Those complexes regulate EMT-associated gene expression.

2)      Please ensure that the diagrams are originally made by you and are not taken from elsewhere. Also, provide the name of the software that was used to make the diagrams.

Thank you for helpful comment. For any diagram that was obtained from a source, we included the reference in the figure legend. Our original diagrams are clearly described in the figure legend. In addition, the specific software used to generate the diagrams was also included in all figure legends.

Reviewer 3 Report

Abe et al in this manuscript summarized the functional relationship between PRMT5 and cancer immunology, with the highlights of regulatory mechanisms of PRMT5 in immunosuppressive TME. This is a well-organized and well-written review which comprises sufficient background knowledge and updated discoveries, and the figures were also decently depicted. Overall in my opinion, this review could be processed for the publication in Genes. I listed the following minor points for the authors’ consideration and hopefully they could help to improve the quality of the manuscript.

1 There was a typo in Fig 2-Equilibrium, “INF-γ” should be “IFN-γ”.

2 Fig 4 was quite blurry. Please upload a higher resolution figure during resubmission.

3 It could be a better idea if the section “3-4. PRMT5 as a Cancer Therapy Target” could be moved to section 5 in page 9.

Author Response

Reply to reviewer 3

1 There was a typo in Fig 2-Equilibrium, “INF-γ” should be “IFN-γ”.

Thank you for the helpful comment. We have changed “INF-γ” to “IFN-γ” in Figure 2.

2 Fig 4 was quite blurry. Please upload a higher resolution figure during resubmission.

Thank you for pointing out the figure resolution issue. We have replaced this with a higher resolution figure.

3 It could be a better idea if the section “3-4. PRMT5 as a Cancer Therapy Target” could be moved to section 5 in page 9.

Thank you for your valuable advice. We agree with your opinion and have moved this to section 5.
